

# Comparison of gut microbiome in the Chinese mud snail (*Cipangopaludina chinensis*) and the invasive golden apple snail (*Pomacea canaliculata*)

Zihao Zhou[1,2,3], Hongying Wu[3], Dinghong Li[3], Wenlong Zeng[3], Jinlong Huang[1,2,3,4] and Zhengjun Wu[1,2,3]

[1] Key Laboratory of Ecology of Rare and Endangered Species and Environmental Protection (Guangxi Normal University), Ministry of Education, Guilin, Guangxi, China
[2] Guangxi Key Laboratory of Rare and Endangered Animal Ecology, Guangxi Normal University, Guilin, Guangxi, China
[3] Guangxi Key Laboratory of Landscape Resources Conservation and Sustainable Utilization in Lijiang River Basin Institute for Sustainable Development and Innovation, Guangxi Normal University, Guilin, Guangxi, China
[4] College of Life Sciences, Guangxi Normal University, Guilin, Guangxi, China

Corresponding authors
Jinlong Huang, jl_huang@163.com
Zhengjun Wu,
wu_zhengjun@aliyun.com

## ABSTRACT

**Background**. Gut microbiota play a critical role in nutrition absorption and environmental adaptation and can affect the biological characteristics of host animals. The invasive golden apple snail (*Pomacea canaliculata*) and native Chinese mud snail (*Cipangopaludina chinensis*) are two sympatric freshwater snails with similar ecological niche in southern China. However, gut microbiota comparison of interspecies remains unclear. Comparing the difference of gut microbiota between the invasive snail *P. canaliculata* and native snail *C. chinensis* could provide new insight into the invasion mechanism of *P. canaliculata* at the microbial level.

**Methods**. Gut samples from 20 golden apple snails and 20 Chinese mud snails from wild freshwater habitats were collected and isolated. The 16S rRNA gene V3–V4 region of the gut microbiota was analyzed using high throughput Illumina sequencing.

**Results**. The gut microbiota dominantly composed of Proteobacteria, Bacteroidetes, Firmicutes and Epsilonbacteraeota at phylum level in golden apple snail. Only Proteobacteria was the dominant phylum in Chinese mud snail. Alpha diversity analysis (Shannon and Simpson indices) showed there were no significant differences in gut microbial diversity, but relative abundances of the two groups differed significantly ($P < 0.05$). Beta diversity analysis (Bray Curtis and weighted UniFrac distance) showed marked differences in the gut microbiota structure ($P < 0.05$). Unique or high abundance microbial taxa were more abundant in the invasive snail compared to the native form. Functional prediction analysis indicated that the relative abundances of functions differed significantly regarding cofactor prosthetic group electron carrier and vitamin biosynthesis, amino acid biosynthesis, and nucleoside and nucleotide biosynthesis ($P < 0.05$). These results suggest an enhanced potential to adapt to new habitats in the invasive snail.

## INTRODUCTION

In recent years, gut microbiota research has made considerable progress (*Gallo, Farrell & Leydet, 2020b*), highlighting the crucial involvement of gut microbes in many mammalian biological processes, such as nutrition absorption, behavior, and intestinal development (*Chen et al., 2021b*; *Li et al., 2019*; *Lize, McKay & Lewis, 2013*). Continuous interactions between the gut microbiota and host have evolved into a dynamic microbial ecological complex (*Brestoff & Artis, 2013*). Unlike in terrestrial vertebrates, gut microbiota in aquatic animals are more variable (*Ringo et al., 2016*) and sensitive to changing conditions (*Zhang et al., 2020*).

The association between hosts and gut microbiota can contribute to our understanding of longevity, metabolism, development, and physiology of invertebrates (*Lee & Hase, 2014*). In shrimp, gut microbiota have been suggested to affect visible nutrient acquisition and disease incidence, which is a major contributor to population restriction (*Xiong, 2018*). Similarly, gut microbiota in other aquatic invertebrates, such as *Daphnia*, play an important role in improvement of high temperature resistance and antiviral ability (*Akbar et al., 2021*). Cellulolytic bacteria in the giant African snail (*Achatina fulica*) gut play an important role in cellulose decomposition (*Pinheiro et al., 2015*). The gut microbiota is important for organisms to adapt to new environment as well. Previous studies have shown that gut microbiota in invasive species have greater potential to invade the new habitats (*Gallo, Farrell & Leydet, 2020a*; *Qu et al., 2020*). For instance, the snail *Potamopyrgus antipodarum* have greater core microbial taxa as an invasive species (*Bankers et al., 2021*). Therefore, comparing the diversity and structure of the gut microbiota between sympatric invasive and native freshwater species may help clarify the underlying mechanisms related to invasion.

Various studies have shown that the gut microbiota can effectively resist internal pathogen invasion and can promote successful biological invasion into new environments (*Becker, Hill & Butaye, 2021*; *Habineza et al., 2019*; *Zhang et al., 2018*). The golden apple snail (*Pomacea canaliculata*) is listed among the 100 worst invasive alien species worldwide by the International Union for Conservative of Nature and the Invasive Species Specialist Group (*Chen et al., 2011a*). This snail causes serious damage to aquatic crops, as well as to wetland floral diversity and ecosystem functioning (*Carlsson et al., 2004*; *Lowe et al., 2000*). The Chinese mud snail (*Cipangopaludina chinensis*) is a native species and popular aquatic food source in China. Chinese mud snail feeds mainly on plant debris, whereas the golden apple snail has a voracious appetite for vegetation, including rice (*Oryza sativa*) crops (*She et al., 2013*). Both snail species occupy similar freshwater habitats, but with different feeding habits in the wild. Whether the differences in feeding habits are closely related to gut microbiota remains unclear. Gut microbiota are closely related to host growth performance (*Fan et al., 2019*). It has been shown that the gut microbiota of golden apple snails can be impacted by developmental stage, gender, and gut part (*Li et al., 2019*; *Lyra et al., 2018*). However, few studies have explored the differences and abundance of gut microbiota in the two sympatric invasive and native freshwater snails. Additionally, we speculated that the gut microbiota may be important for the invasion and adaptation of invasive snails such as the apple snail *P. canaliculata* to new environments. Therefore, to better understand

the gut microbiota community in the two freshwater snails, and investigate the invasive mechanism of golden apple snail at microbial level, we compared the differences in gut microbiota between the invasive golden apple snail and native Chinese mud snail using 16S rRNA sequencing on the Illumina MiSeq platform. This study is of importance for the understanding of invasion mechanism of *P. canaliculata*.

## MATERIALS AND METHODS

### Sample collection and treatment

A total of 20 Chinese mud snails and 20 golden apple snails were collected from the Xiangsi River (25.0459°N, 110.1128°E) in Huixian village, Guilin city, Guangxi, China in June 2021. This research was approved by the ethics committee of Guangxi Normal University (IACUC-202111-003). All snails were collected from shallow water, and sampled by fishing net. The average shell heights of *P. canaliculata* and *C. chinensis* were 3.8 ± 0.3 cm and 5.3 ± 0.3 cm, respectively. The *P. canaliculata* (PC group) and *C. chinensis* (CC group) snails were maintained in 1 L of ultra-pure water in a 25 °C room for 5–7 days. All snails were fed with Chinese cabbage (*Brassica rapa*). Prior to dissection, the snails were starved for 24 h to minimize the amount of partially digested food in the gut.

The shells of snails were cleaned with 70% ethanol two times and ultra-pure water three times. Shell destruction, dissection, and gut extraction were performed on a clean bench. Due to the small size of snail guts, five snail's guts were pooled as a single sample, to make sure the sample adequacy and experiment accuracy was qualified. All samples were repeated four times. A total of eight samples were named with PC-1, PC-2, PC-3, PC-4 and CC-1, CC-2, CC-3, CC-4, respectively. Samples were stored in 5 ml aseptic centrifuge tubes at −80 °C before DNA extraction.

### 16S rRNA gene sequencing

The gut microbial DNA was extracted using a Fast DNA SPIN Extraction Kit (MP Biomedicals, USA) following the manufacturer's instructions. The isolated DNA was stored at −80 °C until polymerase chain reaction (PCR). The DNA concentration and molecular size were measured using UV spectrophotometry (Eppendorf, BioPhotometer, Germany) and 0.8% agarose gel electrophoresis, respectively. The hypervariable V3–V4 region of the 16S rRNA gene was amplified by PCR using universal bacterial primers (338F: 5′-ACTCCTACGGGAGGGAGCA-3′, 806R: 5′-GGACTACHVGGGTWTCTAAT-3′). Each primer included barcode sequences to promote the sequencing of products. The DNA (20 ng) of each sample was amplified in 25-μl reactions consisting of 0.25 μl of Q5 high-fidelity DNA polymerase (NEB, Ipswich, UK), 5 μl of 5-fold reaction buffer, 5 μl of 5-fold high GC buffer, 0.5 μl of dNTP Mix (10 mM), 2 μl of template DNA, 1 μl of each primer (10 mM), and 10.25 μl of ddH$_2$O. The PCR conditions were as follows: an initial denaturation at 98 °C for 5 min, 25 cycles of 98 °C denaturation for 10 s, 50 °C annealing for 30 s, 72 °C extension for 30 s, and a final extension at 72 °C for 5 min. The PCR products were detected by 2% agarose gel electrophoresis and purified with an AxyPrep DNA Gel Extraction Kit (Axygen, New York, Union City, CA, USA). Quantification of PCR products was performed using a Quant-iT PicoGreen dsDNA Assay Kit (Invitrogen, Waltham, MA,

USA). A TruSeq Nano DNA LT Library Prep Kit (Illumina, USA) was used to establish the DNA library. The library was sequenced using a MiSeq Reagent Kit v3 (6,000-cycles-PE) (Illumina, USA) on the MiSeq platform by Personal Biotechnology Co., Ltd. (Shanghai, China).

## Data and analysis

The DADA2 method was used for quality control (*Callahan et al., 2016*). QIIME2 (version: 2019.4) software was used to remove primer sequences, sequences shorter than 150 bp, and chimera sequences. The obtained high-quality sequences were clustered into operational taxonomic units (OTUs) at 97% identity using VSEARCH software (v2.13.4) (*Torbjørn et al., 2016*). The OTUs were annotated using the 16S rRNA database tool (SILVA v132) in QIIME2 (version: 2019.4). The lowest sequence number of OTU abundance was standardized for further study. Raw sequencing reads were submitted to the National Center for Biotechnology Information (NCBI) BioProject database (PRJNA756881).

## Alpha and beta diversity estimation

The alpha and beta diversity indices represent the diversity of species within and between biotopes, respectively (*Whittaker, 1972*). QIIME2 (version: 2019.4) was used to calculate the alpha indexes including community richness (Chao1 and Observed species), diversity (Shannon and Simpson), evolutionary population diversity (Faith's PD), evenness (Pielou's evenness). The calculation methods can be obtained at http://scikit-bio.org/docs/latest/generated/skbio.diversity.alpha.html#module-skbio.diversity.alpha. We used R software to sketch box plots of estimators with an intuitive form. Significance between groups was tested using the Wilcoxon rank-sum and Dunn tests.

To estimate beta diversity and similarity, differences in relative abundance of OTUs between two groups were detected using the nonmetric multidimensional scaling (NMDS). Analysis of similarities (ANOSIM) based on Bray–Curtis distance and weighted UniFrac distance with 999 permutations was used to determine similarities between two groups. Differences in bacterial abundances (from phylum to family) between the two species were tested using linear discriminant analysis effect size (LEfSe).

## Functional prediction and statistical analysis of gut microbiota

Phylogenetic Investigation of Communities by Reconstruction of Unobserved States 2 (PICRUSt2) software (https://github.com/picrust/picrust2/wiki) was used to predict community functions of the gut microbiota (*Douglas et al., 2020*). Greengenes ID corresponding to each OTU was searched in the Kyoto Encyclopedia of Genes and Genomes (KEGG) database and was classified to the relevant KEGG pathway. Numbers of functional genes in each pathway were calculated to compare functional enrichment in gut microbiota between the two groups. Statistical analyses were conducted using independent-samples $t$-test in SPSS (v19.0) at a significance level of $P = 0.05$.

## RESULTS

### Gut microbial diversity and composition in *P. canaliculata* and *C. chinensis*

A total of 737,690 high-quality sequences were obtained from all eight snail DNA samples (PC 1-4, and CC 1-4) after quality control and filtration, including 391,551 reads from the PC group and 346,139 reads from the CC group (ranging from 46,498–187,205 reads per sample). We re-sampled all samples based on the smallest number of reads (46,498) to correct for differences in read number. The rarefaction curve of species based on the Shannon index reached asymptote, indicating that sequencing depth basically covered all species in the samples (Fig. S1).

Filtered sequences were clustered into OTUs at a 97% identity, and we obtained 5,911 valid OTUs after removing those with relative abundances <0.001%, including 4,927 of *P. canaliculata* and 1,080 of *C. chinensis*. Comparing these OTUs with the SILVA database, 6,007 OTUs were annotated to 29 phyla, 52 classes, 123 orders, 247 families, 575 genera, and 963 species. For the PC group, the 4,927 OTUs were annotated to 27 phyla, 49 classes, 111 orders, 239 families, 553 genera, and 894 species. For the CC group, the 1,080 OTUs were annotated to 12 phyla, 18 classes, 44 orders, 67 families, 115 genera, and 150 species (Fig. S2). In addition, the Venn diagram indicated that 96 OTUs existed in both groups. Compared to the CC group (984), however, the number of specific OTUs was much higher in the PC group (4,831) (Fig. S3).

Based on alpha diversity analysis, the Shannon and Simpson indices indicated that gut microbiota diversity did not differ significantly between the two groups. However, the Chao1 and Observed species indices indicated that gut microbiota abundance was significantly higher in the PC group than in the CC group ($P < 0.05$) (Fig. 1).

Differences in beta diversity were evaluated using NMDS analysis based on Bray–Curtis and weighted UniFrac distances. Results showed that the PC and CC groups could be isolated from each other, and intergroup distance was higher than intragroup distance based on both calculation methods (Figs. 2A and 2B). Moreover, ANOSIM revealed significant differences between the PC and CC groups (Bray-Curtis $R = 0.396$, $P = 0.027$, NMDS stress $= 0.0000982$; weighted UniFrac $R = 1$, $P = 0.03$, NMDS stress $= 0.0000776$), as shown in Figs. 2C and 2D. Similar results were observed from principal coordinate analysis (PCoA) based on Bray–Curtis and weighted UniFrac distances (Fig. S4).

### Taxonomic composition of gut microbiome in *P. canaliculata* and *C. chinensis*

From all eight samples, the top 10 gut microbiota phyla were shown in Fig. 3 and Table S1. Comparing bacterial composition between the PC and CC groups, microbes with a relative abundance >2% were defined as dominant phyla. Results showed that Proteobacteria was the dominant phylum in both groups, accounting for 53.78% ± 15.59% in the PC group and 98.94% ± 0.20% in the CC group. In the PC group, three other phyla were found with a relative abundance >2%, including Bacteroidetes (23.19% ± 11.38%), Firmicutes (16.00% ± 6.91%), and Epsilonbacteraeota (4.28% ± 2.53%). For the CC group, Proteobacteria was

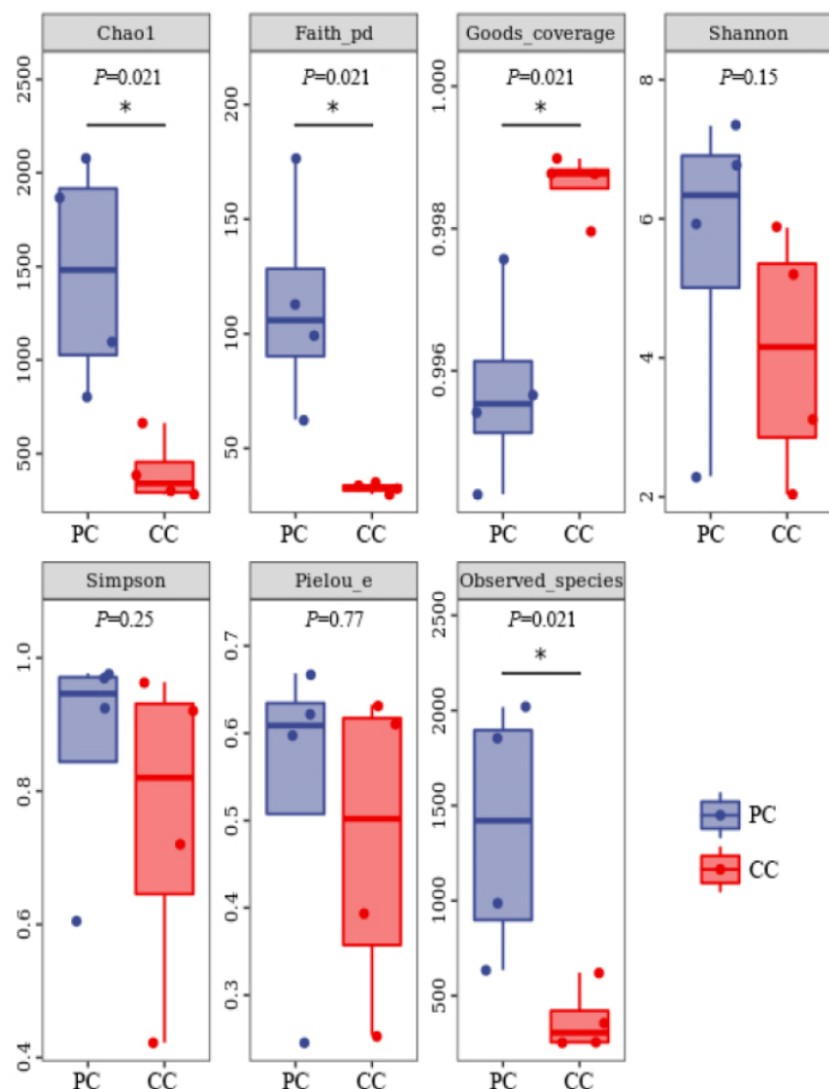

**Figure 1  Alpha diversity index summary.** Chao 1 and Observed species indexes represent community richness; Shannon and Simpson represent diversity; Faith's PD represent the diversity of evolution; Pielou's evenness represent evenness; Good's coverage represent coverage. * $P < 0.05$ represent significant level.

the only phylum with a relative abundance >2%. Of the top 10 phyla, only seven existed in the CC group, with Spirochaetes, Tenericutes, and Fibrobacteres not detected.

At the genus level, the top 10 genera were shown in Fig. 4 and Table S2 . In the PC group, the dominant genus was *Chryseobacterium* (14.62% ± 13.24%), followed by *Lactococcus* (9.78% ± 8.18%), *Uliginosibacterium* (9.73% ± 6.16%), *Aquitalea* (6.13% ± 3.13%), *Novispirillum* (4.11% ± 2.39%), *Sulfurospirillum* (3.50% ± 2.24%), *Dechloromonas* (2.46% ± 2.03%), and *Bacteroides* (2.17% ± 1.95%) (Fig. 4 and Table S2). In the CC group, the dominant genera were *Enterobacter* (6.73% ± 2.31%) and *Hafnia-Obesumbacterium* (2.49% ± 2.18%). Only four of the top 10 genera were found in the CC group, with

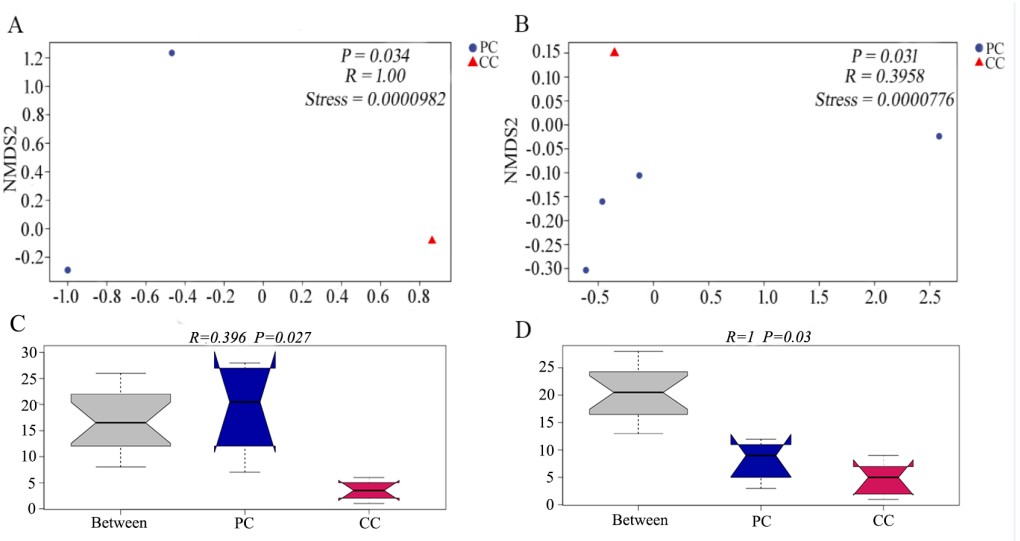

**Figure 2** NMDS and ANOSIM analysis based on (A, C) Bray–Curtis and (B, D) weighted UniFrac distances of gut microbiota on OTU level. The NMDS ordination revealed a significant difference based on Bray–Curtis ($P = 0.034$) and on weighted UniFrac distances ($P = 0.031$) between PC and CC groups. The ANOSIM revealed a significant difference based on Bray–Curtis ($P = 0.027$) and on weighted UniFrac distances ($P = 0.03$) between PC and CC groups. Part of the dots have overlapped (A and B).

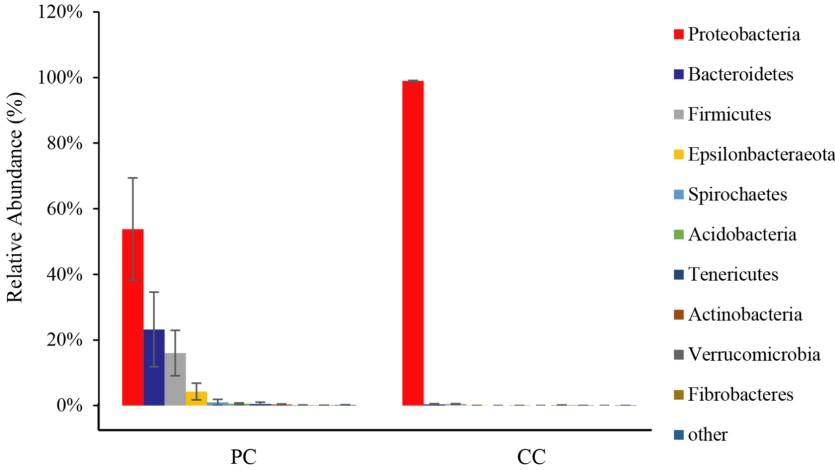

**Figure 3** Relative abundance of top 10 OTU's for *P. canaliculata* (PC) and *C. chinensis* (CC) at the phylum level.

*Chryseobacterium*, *Uliginosibacterium*, *Aquitalea*, *Novispirillum*, *Sulfurospirillum* and *Dechloromonas* not detected. In addition, based on LEfSe analysis (LDA > 2), 10 phyla and 87 genera were identified showing significant differences (Fig. S5).
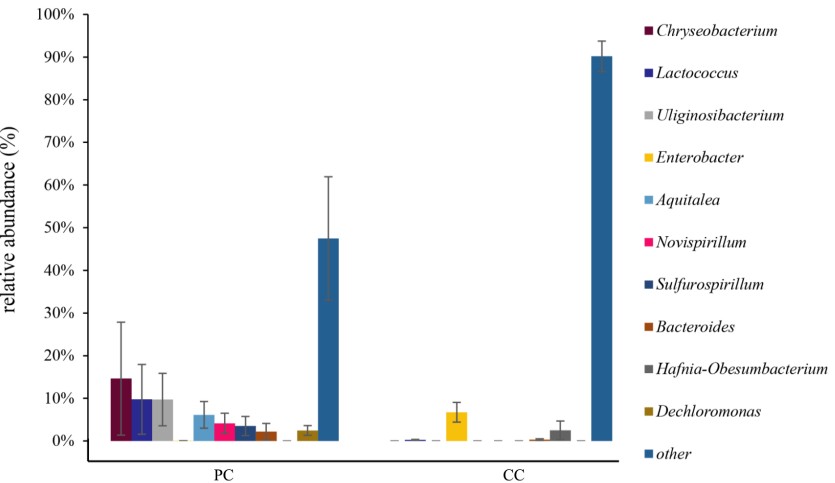

**Figure 4** Relative abundance of top 10 OTU's for *P. canaliculata* (PC) and *C. chinensis* (CC) at the genus level.

## Functional prediction of gut microbiota

Gut microbiota functions were predicted using PICRUSt2. From the KEGG database, a total of 7,186 genes were classified into seven level-1 pathways and 60 level-2 pathways. Of these 60 pathways, 17 were involved in generation of precursor metabolite energy, 15 were involved in degradation/utilization/assimilation, 12 were involved in biosynthesis, 10 were involved in metabolic clusters, two were involved in macromolecule modification, two were involved in glycan pathways, and two were involved in detoxification (Fig. 5). From the level-2 results, cofactor, prosthetic group, electron carrier, and vitamin biosynthesis (16.28% ± 0.38%), amino acid biosynthesis (13.42% ± 0.11%), fatty acid and lipid biosynthesis (9.43% ± 0.60%), and nucleoside and nucleotide biosynthesis (11.53% ± 0.16%) were the most abundant functions (Fig. 5). Among the top abundant functions, cofactor prosthetic group electron carrier and vitamin biosynthesis ($t = -2.861$, $P = 0.029$), amino acid biosynthesis ($t = 17.057$, $P < 0.01$), nucleoside and nucleotide biosynthesis ($t = 10.457$, $P < 0.01$), fatty acid and lipid biosynthesis ($t = 2.887$, $P = 0.028$), carboxylate degradation ($t = -13.277$, $P < 0.01$), secondary metabolite degradation ($t = -10.064$, $P < 0.01$), carbohydrate degradation ($t = -12.963$, $P < 0.01$), nucleoside and nucleotide degradation ($t = 2.533$, $P = 0.044$), aromatic compound degradation ($t = -3.500$, $P = 0.013$), glycolysis ($t = 3.540$, $P = 0.012$), amino acid degradation ($t = -3.725$, $P = 0.010$), C1 compound utilization and assimilation ($t = 4.288$, $P = 0.005$), and aromatic compound biosynthesis ($t = 22.984$, $P < 0.01$) were significantly different between the PC and CC group (Fig. S6).

## DISCUSSION

The composition and diversity of gut microbiota can be affected by a variety of factors, such as the environment, food sources and species (*Bibbo et al., 2016*; *Jin et al., 2017*). Previous study on the gut microbiota of freshwater *Radix auricularia* and *Planorbella trivolvis* snails

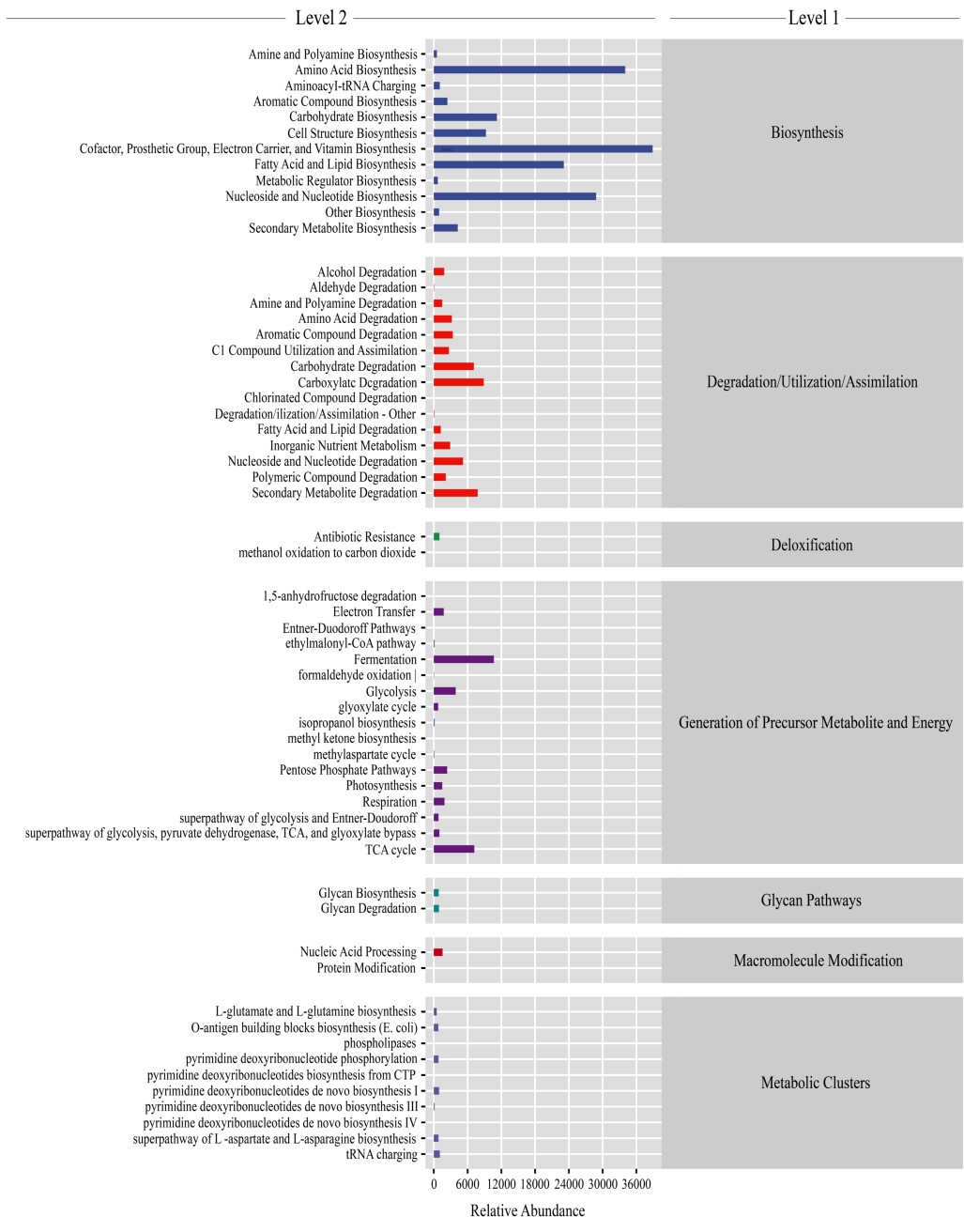

**Figure 5 Gut microbiota predictive metabolic functions from KEGG database in all samples.** The color-bar indicates gene function (level 1). Detailed descriptions are shown on the left side (level 2). The lines at the top of the figure have shown the KEGG level as well.

found that Proteobacteria is dominant in both, but microbial diversity differs significantly, which may be due to disparate sampling sites (*Hu et al., 2018*). In contrast, a six-month study found no significant differences in alpha diversity between invasive red-eared slider turtles (*Trachemys scripta elegans*) and native Chinese three-keeled pond turtles (*Chinemys reevesii*) (*Qu et al., 2020*). In addition, environment plays an important role to the gut

microbiota of the same invasive species as well. It was reported that there were significant differences in the gut microbial community structure of the invasive fish *Oreochromis mossambicus* from the different habitats (*Gaikwad, Shouche & Gade, 2017*). A recent study also showed that the core microbial taxa of invasive freshwater snail *P. antipodarum* in invaded habitat were larger than that in origin habitat (*Bankers et al., 2021*). These studies suggest that the growth environment is a vital factor for gut microbial diversity. In the current study, we found no significant differences in gut microbiota diversity between the two freshwater snails, which were likely due to their similar habitats. However, the Chao1 and Observed species indices showed marked differences in gut microbiota abundance between the two groups, with significantly higher abundance in the PC group than in the CC group. Previous research has found that dietary changes are not significantly correlated with alpha diversity but are positively correlated with beta diversity (*Li et al., 2016*). The beta diversity in this study was significantly different between golden apple snails and Chinese mud snails. Golden apple snails have a more extensive diet than Chinese mud snails (*Morrison & Hay, 2011*), which may be a dominant factor related to the significant differences in beta diversity between the two species. Furthermore, the gut microbiota structure was more complex in the PC group than in the CC group. It was suggested that more complex microbial structure may be able to support the golden apple snail to adapt to the new habitats more rapidly and flexibly.

In this study, the top three phyla in the two groups were Proteobacteria, Bacteroidetes, and Firmicutes, followed by Epsilonbacteraeota, Spirochaetes, and Acidobacteria. Comparing our results with other studies, we found that dominant microbiota were quite similar at the phylum level (*Chen et al., 2021a*; *Li et al., 2019*). We compared results from two relevant studies and found five phyla (Proteobacteria, Bacteroidetes, Tenericutes, Actinobacteria, and Firmicutes) in common (*Chen et al., 2021a*; *Li et al., 2019*). However, Epsilonbacteraeota and Acidobacteria were only found in our study (*Chen et al., 2021a*; *Li et al., 2019*). Epsilonbacteraeota is widely known for clinical relevance and chemolithotrophy and is usually found in sulfide-rich sediments (*Waite et al., 2017*). Acidobacteria is one of the most abundant soil bacteria with a unique ecological function and is found in a wide variety of environments, including under extreme and polluted conditions (*Liu et al., 2017*; *Navarrete et al., 2015*). Epsilonbacteraeota and Acidobacteria were only found in our study, which may be due to environmental differences or differences in long-term food sources of the golden apple snails. Further relevant environmental experiments are needed to screen for factors that may contribute to this phenomenon. Moreover, based on other studies on freshwater snails (e.g., *Radix auricularia*, *Planorbella trivolvis*, and *Bulinus africanus*), the compositions of dominant gut microbiota are very similar at the phylum level (*Hu et al., 2018*; *Hu et al., 2021*; *Van Horn et al., 2012*). Although intestinal microbiota diversity is found in disparate snails, there appear to be core microbes shared among the above-mentioned snails.

Proteobacteria was the dominant microbiota in both groups, and primary microbiota in other Gastropoda like *Potamopyrgus antipodarum* and *Achatina fulica* (*Cardoso et al., 2012b*; *Takacs-Vesbach et al., 2016*). Moreover, Proteobacteria is related to environmental adaptation due to its ability to secrete lipase, protease, and amylase (*Pemberton, Kidd &*

*Schmidt, 1997*). Our results also showed that the Proteobacteria genus *Enterobacter* was dominant at the genus level. *Enterobacter* is considered a cellulolytic genus (*Chen et al., 2021a*; *Pawar et al., 2015*), with certain members, such as the *Enterobacter* sp. strain Bisph2, able to degrade glyphosate (an organic phosphine herbicide) (*Benslama & Boulahrouf, 2016*), and others, such as *Enterobacter* spH1, able to degrade glucose and glycerin into value-added products (*Gueell et al., 2015*). Chinese mud snails live in paddy fields, lakes, and rivers (*Nakanishi et al., 2014*) and primarily feed on diatoms and plant debris (*Cui, Shan & Tang, 2012*), with no reports of carnivorous behavior has been reported yet. Proteobacteria can help in diatom and plant debris digestion. Therefore, we speculate that the relatively simple ''lifestyle'' of Chinese mud snails does not require a complex gut microbiota composition.

In contrast, the diets of golden apple snails are more complex and include plants and aquatic invertebrates, such as *Biomphalaria peregrina*, *Biomphalaria straminea*, and *Physa acuta* (*Cazzaniga, 1990*; *Kwong, Chan & Qiu, 2009*). More complicated diets require more digestion-related bacteria. In addition to digestion-related Proteobacteria, both Bacteroidetes and Firmicutes were related to nutrient absorption (*Sommer et al., 2016*; *Thomas et al., 2011*). Bacteroidetes can degrade high molecular weight organic matter, while Firmicutes was related to the degradation of lipids and dietary fiber (*Chen et al., 2011b*; *Evans et al., 2011*). Both these were highly abundant in golden apple snails, indicating that they had more effective material decomposition and energy absorption than the Chinese mud snail. The functions of these two phyla have also been explored in invertebrates (*Wang et al., 2020b*). In the current study, average Bacteroidetes and Firmicutes abundances were significantly higher in the PC group (23.20% and 16.00%, respectively) than in the CC group (0.34% and 0.49%, respectively). The *Chryseobacterium* genus was only found in the PC group, and the relative abundance of *Lactococcus* was significantly higher in the PC group than in the CC group. *Chryseobacterium* can decompose lignocellulose (*Carlos, Fan & Currie, 2018*; *Puentes Tellez & Salles, 2018*; *Weiss et al., 2021*). In addition, study on the American cockroach (*Periplaneta americana*) reported that *Chryseobacterium* is only found in a high-fiber diet, indicating that *Chryseobacterium* can decompose fiber (*Dugas et al., 2001*). As gram-positive bacteria, *Lactococcus* species are recognized as safe microorganisms for food production and can produce L-lactic acid through acidification to provide energy (*Casalta & Montel, 2008*). Unique or high abundance microbial taxa were related to the material decomposition and capacity utilization, which indicated that the golden apple snail can digest more substances to be more successful at invasion. Therefore, the higher relative abundance and complex composition of gut microbiota in *P. canaliculata* can be one of the reasons for its high survival and adaptability, and thus its successful invasion.

Among the top 10 dominant phyla, three existed in the PC group only: i.e., Spirochaetes, Fibrobacteres, and Tenericutes. In recent years, less attention has been paid to the function of spirochetes, with most studies limited to pathogens involved in Lyme disease, recurrent fever, and syphilis (*Gattorno et al., 2019*; *Hook III, 2017*; *Radolf et al., 2012*). In the current study, we speculate that the Spirochaetes bacteria are parasites rather than functional bacteria in the golden apple snails. Fibrobacteres is a well-known primary degrader of cellulose in the intestinal tract of herbivores and can hydrolyze polymer using a distinctive

set of glycoside hydrolases and binding domains (*Rahman et al., 2016*). Tenericutes is suggested to be involved in carbohydrate storage, carbon fixation, and environmental response (*Wang et al., 2020a*). In this study, both Fibrobacteres and Tenericutes were unique dominant microbiota in golden apple snail, which may be related to their high invasiveness.

Functional KEGG predictions indicated that many functions were significantly different between the PC and CC groups, including cofactor, prosthetic group, electron carrier, vitamin biosynthesis, amino acid biosynthesis, nucleoside and nucleotide biosynthesis, fatty acid and lipid biosynthesis, and carboxylate degradation, consistent with previous studies on indigenous species *Helix pomatia* and invasive species *Achatina fulica* (*Cardoso et al., 2012a*; *Nicolai et al., 2015*). Of these functions, the relative abundance in the PC group was significantly higher than that in the CC group, including amino acid biosynthesis, fatty acid and lipid biosynthesis, aromatic compound biosynthesis, and C1 compound utilization and assimilation. Gut microbiota are crucial to host amino acid homeostasis and health (*Mardinoglu et al., 2015*). Several genera are known to play crucial roles in amino acid biosynthesis, including *Fusobacterium*, *Bacteroides*, and *Veillonella* (*Lin et al., 2017*). Additionally, *Lactococcus* and *Bacteroides* are associated with fatty acid and lipid biosynthesis (*Liu & Meng, 2008*; *Tanca et al., 2018*). In this study, the relative abundances of *Bacteroides* and *Lactococcus* were significantly higher in the PC group and are suggested to be the main factors for the biosynthesis of basic elements. In addition to fatty acid and lipid biosynthesis, *Lactococcus*, a dominant genus in our study, also produces exopolysaccharide and aromatic compounds (*Casalta & Montel, 2008*). Our results indicated that the two species showed considerable differences in gut microbiota functions. The higher gene functions in PC group were focused on nutrient utilization, indicating that the gut microbiota in golden apple snail has higher nutrient conversion capacity than the Chinese mud snail, which helps the golden apple snail to improve the survivability in unstable environment.

## CONCLUSIONS

Our study provided a general understanding of the gut microbial community characteristics between sympatric invasive golden apple snail and native Chinese mud snail. We found that microbial abundance and community structure was significantly different between golden apple snail and Chinese mud snail. The abundance and community structure of gut microbiota in golden apple snail was higher and more complex than in Chinese mud snail, suggesting that the golden apple snail may have the ability to digest a wider variety of food. In addition, we found some specific bacteria phyla and genera of golden apple snail gut microbiota, which can promote the ability to absorb and transform nutrients helping golden apple snail to be more invasive. These results revealed that the structure and function in microbiota composition are the reasons why golden apple snail readily adapts to the new habitats.

## ACKNOWLEDGEMENTS

We thank Lina Du for species identification. We thank Jie Zhu and Jiali Wei for help with feeding snails.

### Funding

This study was supported by the National Natural Science Foundation of China (No. 31960280), Guangxi Natural Science Foundation of China (No. 2018GXNSFAA294024), Guangxi Science and Technology Base and Talent Project (No. AD20159042), Key Scientific Research Program of Guangxi Normal University (No. 2017ZD009) and Guangxi Universities Foundation Promotion Project (No. 2019KY0055). The funders had no role in study design, data collection and analysis, decision to publish, or preparation of the manuscript.

### Grant Disclosures

The following grant information was disclosed by the authors:
National Natural Science Foundation of China: 31960280.
Guangxi Natural Science Foundation of China: 2018GXNSFAA294024.
Guangxi Science and Technology Base and Talent Project: AD20159042.
Key Scientific Research Program of Guangxi Normal University: 2017ZD009.
Guangxi Universities Foundation Promotion Project: No. 2019KY0055.

### Competing Interests

The authors declare there are no competing interests.

### Author Contributions

- Zihao Zhou conceived and designed the experiments, performed the experiments, analyzed the data, prepared figures and/or tables, authored or reviewed drafts of the paper, and approved the final draft.
- Hongying Wu performed the experiments, analyzed the data, prepared figures and/or tables, and approved the final draft.
- Dinghong Li and Wenlong Zeng analyzed the data, authored or reviewed drafts of the paper, and approved the final draft.
- Jinlong Huang and Zhengjun Wu conceived and designed the experiments, analyzed the data, prepared figures and/or tables, authored or reviewed drafts of the paper, and approved the final draft.

### Field Study Permissions

The following information was supplied relating to field study approvals (i.e., approving body and any reference numbers):

Field experiments were approved by the ethics committee of Guangxi Normal University (project number: 202111-003).

## Data Availability

The data is available at the National Center for Biotechnology Information: PRJNA756881.

## Supplemental Information

Supplemental information for this article can be found online at http://dx.doi.org/10.7717/peerj.13245#supplemental-information.

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
