# Peer review of "Comparison of gut microbiome in the Chinese mud snail (Cipangopaludina chinensis) and the invasive golden apple snail (Pomacea canaliculata)"

_PeerJ, doi:10.7717/peerj.13245_

## Round 0.1 · original submission · Minor Revisions

I have now received two reviews for your report. Both agree that this study is worthy of publication following revision. A part of the the 1st point raised by Reviewer 1 is not factually correct (you already have a structured abstract). Please consider your revision based on the other comments raised by the two reviewers. My main concern - aim to provide a clear objective (hypothesis) and a conclusion for your paper.

·

Basic reporting

I commend all authors for their study conducted. However, I recommend major revision for this article before getting accepted. Following are the suggestions that the authors need to follow prior acceptance of this manuscript.
1. Abstract needs to be rewritten and structured with subheadings.
2. Introduction: Do you have any record, of when did the other snail species invasion took occurred. If so, make a mention of the same here in the introduction part.
3. Materials and Methods: How were the snails collected? Objective of the study needs to be very clear and you need to mention hypothesis.
4. Results need to be re-organised and clear. It would be nice if you could provide a table of percentage of phyla and so on, for the species percentage composition for the two species.
5. Discussion needs to be revised. You need to put forward your explanation of how would the invasion mechanisms would be traced through gut microbiome, as this is also one of your objective of study. You can consider the following flow for writing discussion.
6. Conclusion drawn from the study conducted and the object of the study are not corelated in the manuscript. Conclusions are not well stated.
7. Journal Format is poorly followed in the reference section.
 The reference section needs to be revised according to the journal format. The references need to be in chronological order.
 Never copy and paste the references.
 The font size should be 12 instead of 10.5 throughout the reference section.
 Genus and species need to be in italics and not normal fonts even in the reference section.
 Provide DOI number for every article cited, if available.
All figure legends are poorly done, which hardly provide information of what was done in which species. Elaborate the figure legends thoroughly.
8. Throughout the manuscript insert comma after et al.
9. All figure should have same font styles.
There is no mention of ethical permission in the article which needs to be added.
10. Either use symbol or words for “alpha” throughout the manuscript that includes text, figures, figure legends and tables.
11. For the statistical analysis, “t-test” “t” needs to be in italics. Replace “t” with “t” throughout manuscript

Experimental design

The study design is simple and can be replicated, but the method to collect the individuals is missing. There is no mention of ethical permission in the article which needs to be added. Statistical analysis is done adequately and thus the results are well supported for the tests conducted.

Validity of the findings

Results need to be re-organised and clear. It would be nice if you could provide a table of percentage of phyla and so on, for the species percentage composition for the two species. Discussion needs to be revised. You need to put forward your explanation of how would the invasion mechanisms would be traced through gut microbiome, as this is also one of your objective of study. You can consider the following flow for writing discussion. Conclusion drawn from the study conducted and the object of the study are not correlated in the manuscript. Conclusions are not well stated.

Additional comments

Line No 1-4: Kindly modify title of your manuscript as it is not appealing.
Line No 45-80: Rewrite the introduction. Write the objective of study and hypothesis clearly.
Line No 82-89: You need to mention, how were the snails collected.
Line No 90-93: Why did you mix the digestive system of five snails together and grouped them? Was the sample of each snail gut not sufficient for the gut microbiome analysis?
Line No 106-108: PCR conditions-mention details (rewrite).
Line No 173: What are those 8 samples? You need to mention it.
Line No 173-177: Rewrite. Each group should have its own percentage.
Line No 178: Which two groups??
Line No 186-199: Revise. You should be clear, which group of genus has the respective percentage. It would be more clear, if you could represent the same in the form of table.
Line No 209: Replace “Function prediction of gut microbiota” with “Functional prediction of gut microbiota”.
Line No 232: “The composition and diversity of gut microbiota can be affected by a variety of factors.” Which factors? You also need to mention about the invasion of species and then the differences of native and invasive species.
Line No 243-245: “However, the Chao1 and Observed species indices showed marked differences in gut microbiota abundance between the two groups, with significantly higher abundance in the PC group than in the CC group”. What do you mean here?
Line No 342-347: Conclusion is not well stated and needs to be revised.
Line No 489-492: Change from capitals to small letters.
Figure Legends: Revise the figure legends.
 Figure 1: Here you need to mention, which Alpha diversity indexes have been investigated.
 Figure 4: Revise figure legends. You need to mention the significance values here and add additional details.
 Figure 5: Revise figure legends.
 Figure S1: “Rarefaction Curve based on Shannon index”. Add more details, which species and how many replicates were used.
 Figure S2: Revise figure legends.
 Figure S3: “Venn diagram of shared and unique OTUs among different groups”. You need to mention here, which were the different groups examined.
 Figure S4: Revise figure legends.
 Figure S5: Revise figure legends.
 Figure S6: Revise figure legends.

·

Basic reporting

Comments to the Authors
Review of "Comparison and diversity of gut microbiota between invasive golden apple snail (Pomacea canaliculata) and native Chinese mud snail (Cipangopaludina chinensis) (#68660)"

In this study, the authors compared gut microbiota of two sympatric freshwater snails (invasive golden apple snail and native Chinese mud snail). It is an interesting study for helping clarify invasion mechanism of golden apple snail at microbial level, with comparation from local snail species. However, there still are some questions below. I suggest it would be considered acceptation for publication at PeerJ after addressing the following comments.

Experimental design

The experimental design of this study is good, and the experiment was carried out very well.

Validity of the findings

This paper looks at the interesting comparation of gut microbiota between invasive snail species and local snail species. The results are very interesting, which can clarify invasion mechanism of golden apple snail at microbial level.

Additional comments

Comments to the Authors
Review of "Comparison and diversity of gut microbiota between invasive golden apple snail (Pomacea canaliculata) and native Chinese mud snail (Cipangopaludina chinensis) (#68660)"

In this study, the authors compared gut microbiota of two sympatric freshwater snails (invasive golden apple snail and native Chinese mud snail). It is an interesting study for helping clarify invasion mechanism of golden apple snail at microbial level, with comparation from local snail species. However, there still are some questions below. I suggest it would be considered acceptation for publication at PeerJ after addressing the following comments.

Major and minor comments:
Introduction
1) Insufficient focus in the introduction. For example, line 53-59: The authors describe the importance of gut microbes to hosts. However, they did not support the statement "Therefore, comparing the diversity and structure of the gut microbiota between sympatric invasive and native freshwater species may help clarify the underlying mechanisms related to invasion. (line 59-61)". Please provide some references for the statement and reworded the paragraph.
Materials and Methods
1) Line 89-92: The authors used 20 Chinese mud snails and 20 golden apple snails. Why did you mix five snails as a single sample? Please provide more detailed sample collection information.
Results
1) Line 215-223: There are F value and t value with one p value. Why does a significant difference have both F and T values? Please illustrate and correct.
2) Figure 4(A and B): CC group has only one dot in NMDS plot (A and B). Are they overlapped? Please illustrate.
3) Figures are too fuzzy. For example figure 4 and 5. Please provide high quality figure.
Discussion
1) Line 249: "golden apple snail" inconsistencies in the text
2) Line 288-291: "In addition, the ratio of these two phyla can have a significant influence on energy absorption in rats." Does high ratio promote energy absorption? Please make it clear to support your result.
3) Line 284-285: "Bacteroidetes functions in the degradation of high molecular weight organic matter …" It needs editing for grammar and sentence structure for this sentence.

---

## Round 0.2 · Minor Revisions

You have addressed many of the concerns or the reviewers, but some remain. One of the reviewers have several suggestions for improvement that you should consider in your next revision. Though I agree with the idea that the title needs to be slightly more descriptive, the suggestions by the reviewer are also not ideal. Please bear this in mind when making the change.

·

Basic reporting

Initially, I commend all authors for their study conducted. However, I recommend “Minor Revision” for this article before getting accepted. Following are the suggestions that the authors need to follow on prior acceptance of this manuscript.
1. Title “An insight into the role of gut microbiota in successful invasion of golden apple snail (Pomacea canaliculata)” is not clear and needs to be revised. I suggest you to change the title. You can revise it for example (a) “Investigation of snail gut microbiome between the native (Cipangopaludina chinensis) and the invaded (Pomacea canaliculata) species. (b) “Comparison of the gut microbiome in the native (Cipangopaludina chinensis) and the invaded snail species (Pomacea canaliculata) in China” etc.
2. Abstract: Include the objective of your study.
3. Introduction: You need to mention, of what is your hypothesis for the present study conducted. Also, the objective of the study conducted is not stated clearly. Kindly make a mention at the end of the introduction.
4. Discussion needs to be revised as some information overlaps.
5. Conclusion made from your study conducted, mentions of community and diversity of microbiomes in the two species (native and invaded), but does not state how the gut microbiome helps in understanding the invasion mechanism which needs to be focused.
6. Corrections followed by Journal format need to be done in the text and reference section.
Line Number 11; Line Number 26; Line Number 56; Line number 59; Line Number 62; Line Number 62; Line Number 67; Line Number 70; Line Number 94; Line Number 104; Line Number 107; Line Number 119; Line Number 127; Line Number 131; Line Number 136-139; Line Number 143; Line Number 160; Line Number 165; Line Number 168; Line Number 189; Line Number 191; Line Number 207; Line Number 232; Line Number 239-240; Line Number 241; Line Number 249-252; Line Number 430; Line Number 460; Line Number 483; Line Number 486; Line Number 490; Line Number 499; Line Number 514; Line Number 547; Line Number 553; Line Number 598; Line number 624; Line Number 628; Line Number 630.

Experimental design

The study conducted is simple and can be replicated. Statistical analysis are conducted appropriately which are in support of the results obtained for the present study.

Validity of the findings

The objective of the study and conclusion are not well stated. How gut microbiome help in understanding the invasion mechanism needs to be focused here.

Additional comments

Line Number 11: “Lijiang RiverBasin Institute” Replace with “Lijiang River Basin Institute”.
Line Number 26: “Methods” this can be a separate section of abstract.
Line Number 56: Delete “-” after high.
Line number 59: Add “to” after adapt.
Line Number 62: Change to italics “Potamopygus antipodarum”.
Line Number 62: Rewrite this “as invades than the native”.
Line Number 67: Add comma after shown that the
Line Number 70: Add reference in text and include the same in reference section “listed as 100 worst invasive alien species”.
Line Number 94: Replace “no damage to the surrounding environment” with “without disturbing the surrounding environment”.
Line Number 104: Replace “PC1, 2, 3, 4 and CC 1, 2, 3, 4” with “PC-1, PC-2. PC-3, PC-4 and CC-1, CC-2, CC-3, CC-4”.
Line Number 107: Change “gut microbiota DNA” to “Gut microbial DNA”
Line Number 119: Delete “and” before 72 C.
Line Number 127: Which version of QIIME2 software was used.
Line Number 131: Which version of QIIME2 software was used.
Line Number 136-139: Rewrite “Alpha diversity indices, including community richness (Chao1 and Observed species), diversity (Shannon and Simpson), diversity of evolution population (Faith’s PD), evenness (Good’s coverage), were calculated using QIIME2”. Also, include which version of QIIME2 was used.
Line Number 143: “To estimate beta diversity or similarity” Replace “or” with “and”.
Line Number 160: Replace the subheading “Diversity and composition of P. canaliculata and C. chinensis gut microbiota” to “Gut microbial diversity and composition in P. canaliculata and C. chinensis”.
Line Number 165: Delete an extra space after species.
Line Number 168: Insert a space after the symbol <0.001%.
Line Number 189: Replace the subheading from “Taxonomic composition of P. canaliculata and C. chinensis gut microbiota” with “Taxonomic composition of gut microbiome in P. canaliculata and C. chinensis”.
Line Number 191: Delete “Tab” before, Comparing.
Line Number 207: Delete “as” after identified.
Line Number 232: Insert space after “species”.
Line Number 239-240: Rewrite the sentence “It was reported that the microbial community structure of invasive fish Oreochromis mossambicus gut microbiota was significantly different from two habitats”.
Line Number 241: Replace “recently” with “recent”.
Line Number 249-252: Do not include the results in discussion, indeed you can write what inferences you draw from your results obtained. Rewrite this section again. In our study, the beta diversity results showed that the intestinal microbial community structure of samples within the same species were highly similar, but there were significant differences in the community structure of samples among different species(P < 0.05)”.
Line Number 430: Change to italics “Frontiers in Veterinary Science”.
Line Number 460: Replace “Peerj” with “PeerJ”.
Line Number 483: Replace “Peerj” with “PeerJ”.
Line Number 486: Replace “Peerj” with “PeerJ”.
Line Number 490: Delete extra “dot” after snails.
Line Number 499: Replace “Bmc” with “BMC”.
Line Number 514: Delete the space before “the World Conservation Union:12 pp”.
Line Number 547: Delete the space after “860”.
Line Number 553: Replace “Peerj” with “PeerJ”.
Line Number 598: Replace “Bmc” with “BMC”.
Line number 624: Add space before and after “<”.
Line Number 628: Add a space before and after “=”.
Line Number 630: Add a space before and after “=”.

·

Basic reporting

The authors almost addressed all the comments from reviewers, and it is fine by me to accept this revised version.

Experimental design

The authors almost addressed all the comments from reviewers, and it is fine by me to accept this revised version.

Validity of the findings

The authors almost addressed all the comments from reviewers, and it is fine by me to accept this revised version.

Additional comments

No additional comments, thanks.

---

## Round 0.3 · Minor Revisions

Your paper is now almost ready to be accepted. However, it still suffers from some language related errors. I have gone through your manuscript and suggest some changes to improve the presentation of your study (see the attached document). Please make these changes and also check for other syntactical errors carefully in your final revision.

---

## Round 0.4 · accepted · Accept

You have now addressed all the reviewer and my final comments to an appreciable degree, meeting the editorial requirements for submission. I am delighted to accept your paper while wishing the best with your future research work.